# Homeostatic Regulation of Glucocorticoid Receptor Activity by Hypoxia-Inducible Factor 1: From Physiology to Clinic

**DOI:** 10.3390/cells10123441

**Published:** 2021-12-07

**Authors:** Davide Marchi, Fredericus J. M. van Eeden

**Affiliations:** 1Department of Biology and Biochemistry, University of Bath, Claverton Down, Bath BA2 7AY, UK; 2The Bateson Centre & The School of Biosciences, Firth Court, University of Sheffield, Western Bank, Sheffield S10 2TN, UK; f.j.vaneeden@sheffield.ac.uk

**Keywords:** glucocorticoid, glucocorticoid receptor, hypoxia inducible factor, crosstalk, immune modulations, inflammation

## Abstract

Glucocorticoids (GCs) represent a well-known class of lipophilic steroid hormones biosynthesised, with a circadian rhythm, by the adrenal glands in humans and by the inter-renal tissue in teleost fish (e.g., zebrafish). GCs play a key role in the regulation of numerous physiological processes, including inflammation, glucose, lipid, protein metabolism and stress response. This is achieved through binding to their cognate receptor, GR, which functions as a ligand-activated transcription factor. Due to their potent anti-inflammatory and immune-suppressive action, synthetic GCs are broadly used for treating pathological disorders that are very often linked to hypoxia (e.g., rheumatoid arthritis, inflammatory, allergic, infectious, and autoimmune diseases, among others) as well as to prevent graft rejections and against immune system malignancies. However, due to the presence of adverse effects and GC resistance their therapeutic benefits are limited in patients chronically treated with steroids. For this reason, understanding how to fine-tune GR activity is crucial in the search for novel therapeutic strategies aimed at reducing GC-related side effects and effectively restoring homeostasis. Recent research has uncovered novel mechanisms that inhibit GR function, thereby causing glucocorticoid resistance, and has produced some surprising new findings. In this review we analyse these mechanisms and focus on the crosstalk between GR and HIF signalling. Indeed, its comprehension may provide new routes to develop novel therapeutic targets for effectively treating immune and inflammatory response and to simultaneously facilitate the development of innovative GCs with a better benefits-risk ratio.

## 1. Introduction

The name “glucocorticoid” (GC) is a portmanteau word (glucose + cortex + steroid), which derives from their key role in the regulation of glucose metabolism, their biosynthesis at the level of the adrenal cortex and their steroidal structure. They represent a well-known class of lipophilic steroid hormones synthetized, with a circadian rhythm, by the adrenal glands in humans and by the inter-renal tissue in teleost fish. GC circadian production in mammals is tuned by the hypothalamus-pituitary-adrenal (HPA) axis, which is the equivalent of the hypothalamus-pituitary-inter-renal (HPI) axis in teleost fish. Both are essential for stress adaptation [1,2,3,4]. The axis consists of a highly conserved regulatory system present in all living organisms aimed at maintaining a dynamic equilibrium in the body in response to external and internal stimuli, which is fundamental to assure homeostasis and survival. Cortisol, the end-product of the HPA/I axis, is the main GC both in humans and teleost fish and plays a fundamental role in the maintenance of both resting and stress-related responses [5,6,7,8].

Since their discovery in the 1940s [9], much has been learnt of GC molecular modes of action [10,11,12,13,14,15,16,17,18,19,20,21,22,23]. In particular, the glucocorticoid receptor’s characterization as a DNA-binding protein that regulates transcription initiation [24], the cloning of GR [25,26] and the breakthrough that most of the immunosuppressive actions of GCs occur via interfering with key inflammatory transcriptional regulators such as NF-κB and AP-1 [27,28,29,30], represent the main milestones.

Natural and synthetic glucocorticoids have been widely used for decades as effective anti-inflammatory and immunosuppressive treatments to control pathological disorders, which are very often linked to hypoxia. In particular, they have been broadly used to treat both acute and chronic inflammations, including inflammatory bowel disease, rheumatoid arthritis, multiple sclerosis, eczema and psoriasis, as well as being used in treatment of various leukaemias and in immunosuppressive regimes upon organ transplant [31,32,33,34,35,36]. At any point in time, an estimated ~1% of the total adult UK population receives oral glucocorticoid therapy [37]. However, due to the presence of adverse effects [38] and GC resistance [39,40,41,42], their therapeutic benefits are limited in patients chronically treated with these steroids. Examples of the most common GC-related side effects include osteoporosis, glaucoma, diabetes, skin atrophy, abdominal obesity, dyslipidemia, hypertension in adults and growth retardation in children [16,43,44].

Cortisol exerts its functions through direct binding to the glucocorticoid receptor (GR), but also to the mineralocorticoid receptor (MR), which binds cortisol with even higher affinity [45,46]. As transcription factors, both GR and MR compete for the same ligands, can form heterodimers and homodimers with each other, recognize and bind many of the same hormone response elements on the DNA, and share numerous coregulatory proteins involved in the gene transcription initiation. Importantly, GCs activate MR in most tissues at basal levels, whereas activate GR under stressful conditions or at the diurnal peak [47].

Once bound together, they form an active complex which can function in the nucleus to modulate the transcription of effector proteins, as well as in the cytoplasm to hamper targeted transcription factors activity. Historically, these functions have been coined genomic and nongenomic modes of action, respectively [48,49,50,51]. Importantly, GCs and their kindred intracellular receptors, represent critical checkpoints in the endocrine control of vertebrate energy homeostasis. Indeed, if HPA axis activity is not accurately regulated, GC imbalance may result in different pathological conditions such as hypertension, severe cardiovascular, immunological and metabolic complications (e.g., Addison’s disease (GC deficiency) and Cushing’s syndrome (GC excess)) [52,53,54]. In addition, alterations or flaws in the HPA axis response are tightly associated with a broad range of inflammatory and autoimmune diseases, both in humans and in animal models. The latter include Crohn’s disease, rheumatoid arthritis, colitis, inflammatory bowel disease, multiple sclerosis (whose animal equivalent is autoimmune encephalomyelitis), dermatitis, and asthma. Inflammatory conditions include fibromyalgia, chronic fatigue syndrome, depression, and post-traumatic stress disorder (PTSD) [55,56,57,58,59,60,61]. Moreover, even if the biological effects induced by GCs are usually adaptive, their abnormal activity may contribute to a series of acute metabolic diseases which include insulin resistance, obesity, and type 2 diabetes [62,63]. Thus, furthering the research on how GCs precisely work and interact with other pathways may provide better tools to treat these diseases and simultaneously allow the development of selective GR agonists and specific drug-targeting strategies.

Similarly to GCs that are involved in numerous homeostatic maintenance activities (e.g., metabolism of protein, carbohydrate and lipid, etc.) [64], the HIF signalling pathway exerts a pivotal role in ensuring homeostasis, the preservation of which is essential for the correct functioning of the cell. In this regard, the ability to perceive and quickly respond to changes related to environmental oxygen availability is controlled by the hypoxia-inducible factor transcription factors (HIF) family. Hypoxia is a common pathophysiological occurrence, with a profound impact both on human and animal physiology, in which oxygen availability to cells, tissues or to an organ is reduced below a certain threshold (O_2_ levels < 2%) [65,66]. HIF transcription factors are key homeostatic regulators which coordinate a metabolic shift from aerobic to anaerobic metabolism to assure cell survival, both in mammals and in zebrafish [67,68,69,70,71].

The HIF pathway is finely regulated by the PHD3-VHL-E3 ubiquitin ligase complex the aim of which is to maintain low basal HIF levels that can rapidly increase to promptly respond when oxygen levels decrease. This avoids any activation of the HIF pathway under normoxic conditions. As a transcription factor, HIF drives the hypoxic response via binding to specific hypoxia-response elements (HREs). These are involved in decreasing oxygen consumption and increasing oxygen and nutrient delivery [72,73,74]. Interestingly, HIF signalling can tune its own activation via negative feedback by inducing the expression of the oxygen sensors proteins (PHDs), in particular prolyl hydroxylase 3 (PHD3) in zebrafish and PHD2 in humans and mice [75,76].

However, although the HIF response is aimed at restoring tissue oxygenation and perfusion, it may sometimes be maladaptive and may contribute to the onset of different pathological conditions (e.g., inflammation, stroke, tissue ischemia and growth of solid tumours) [65]. Thus, both glucocorticoids and hypoxia-induced transcriptional responses have been shown to exert crucial roles in tissue homeostasis and in the regulation of cellular responses to stress and inflammation [77,78,79,80,81,82].

Recent studies have also strengthened the knowledge on the important crosstalk between these two major signalling pathways. To this purpose, the aim of the present review is to discuss the evidence accumulated to date about such crosstalk. We also describe novel mechanisms by which GR and HIF influence each other, both in vitro and in vivo, and how these could be exploited to develop novel therapeutic targets required to overcome GC-related and HIF-related diseases. To this end, the use of the zebrafish (*Danio rerio*) as an effective in vivo model organism has become more and more important to study how both hypoxic and glucocorticoid signalling function in vivo. Indeed, zebrafish share all the components of the human HIF and GC pathway, and the zebrafish has been demonstrated to be a very informative and genetically tractable organism for studying both hypoxia and the stress response both in physiological and pathophysiological conditions [76,83,84,85,86,87]. Finally, the ease use of medium to high throughput drug screening [88] and genetic tractability via the CRISPR/Cas9 based mutagenesis method [89,90,91] make the zebrafish particularly suitable not only for genetic dissection of pathways themselves, but also crosstalk between pathways studied at the level of whole organism.

## 2. Glucocorticoids

### 2.1. Biosynthesis, Secretion and Availability

GCs are essential steroid hormones biosynthesized and secreted by the adrenal cortex/inter-renal gland both in a circadian manner and in response to stress. The latter is generally defined as a status of real or perceived threat to homeostasis. Assuring homeostasis in the presence of stressors requires the activation of an intricate series of coordinated biological responses performed by the nervous, endocrine and immune systems [62,92]. The key anatomical structures that regulate the stress response are located both in the central nervous system and in peripheral tissues. The primary effectors of the stress response are localized in the paraventricular nucleus (PVN) of the hypothalamus, in the anterior lobe of the pituitary gland and at the level of the adrenal gland. These three main structures are generally referred to as the hypothalamic-pituitary-adrenal (HPA) axis in humans, and as the hypothalamic-pituitary-inter-renal axis (HPI) in zebrafish [62,92,93]. Among these, the hypothalamus is the initial stressor recognition site for both internal and external signals. In mammals, neurons localized in the paraventricular nucleus synthesize both corticotropin-releasing factor (CRF) and arginine vasopressin (AVP), which are released into hypophyseal portal vessels that access the anterior pituitary gland. On the other hand, in teleosts, there is a direct neuronal connection to endocrine cells through the hypophyseal stalk, since they lack a portal system between the hypothalamus and the pituitary gland [94]. Here, CRF binding to its receptor localized on pituitary corticotropes triggers the release of adrenocorticotropic hormone (ACTH) into the systemic circulation. In humans, ACTH derives by post translational modification of its precursor encoded by the proopiomelanocortin (POMC) gene. Of note, due to genome duplication, two *pomc* genes named *pomca* and *pomcb* have been identified in zebrafish [95,96,97]. However, only *pomca* seems to be expressed in the pituitary gland and is required for the inter-renal organ development [98,99]. The main target of ACTH is the adrenal cortex in humans and the inter-renal tissue in teleosts, where it binds to the melanocortin 2 receptor (MC2R) on the steroidogenic cells. Here, it stimulates cortisol biosynthesis and secretion starting from cholesterol [62,92,100,101].

Finally, once released into the systemic circulation, GCs can access target tissues (e.g., liver, heart, and vascular tissues) to exert metabolic and cardiovascular effects and the brain itself, in order to support cognitive processes required to tackle a threatening situation [102]. Under non stressful conditions, glucocorticoid levels in the serum are homeostatically controlled by the HPA monitoring activity, whereas glucocorticoid availability is further tuned at a tissue and cellular level. Circulating glucocorticoids are primarily bound to corticosteroid binding globulin (CBG) and just a small percentage (5–15%) is bound to albumin. As a result, the majority of GC is maintained in an inactive form, and only the remaining 5% of systemic GC is free and bioactive. Hence, CBG concentration constitutes a pivotal regulator of cortisol accessibility [103].

Importantly, GCs are also able to control their own biosynthesis and secretion by tuning the activity of the HPA/I axis itself. This is particularly important to stop the stress response and avoid an exacerbated reaction [104]. This is achieved via a GC-GR mediated negative feedback loop, which acts both at the hypothalamic and anterior pituitary levels, where GC-GR activity inhibits both CRH and POMC (ACTH precursor) biosynthesis and release [2,21,23,105]. This occurs via a mechanism that requires GC-GR binding to an nGRE within the *pomca* promoter [106]. For these reasons, *pomca* is a well-established and frequently used readout of GR activity. In addition, GCs may indirectly control the HPA axis activity through modulation of brain structures activity, such as the amygdala, the hippocampus and the prefrontal cortex, that can, in turn, influence the activity of the paraventricular nucleus [102,107,108,109].

### 2.2. The Glucocorticoid Receptor: Structure and Functions

GCs exert their systemic functions by binding to the glucocorticoid receptor (GR) and the mineralocorticoid receptor (MR). Due to their lipophilic nature, GCs can passively diffuse across the plasma membrane into the cytoplasm. Within the cells, their biological availability is then regulated by two enzymes of the 11β-Hydroxysteroid dehydrogenase (11β-HSD) family that work in an opposite fashion. 11β-HSD2 oxidizes cortisol into its inactive form cortisone, reducing GC availability. Vice versa, 11β-HSD1 transforms cortisone to cortisol, thereby increasing local GC activity. Inside the cell, GCs can bind to their specific receptors GR and MR [33,102,110,111]. Both receptors, in the absence of their ligands (unbound state) are associated in an inactive oligomeric complex with specific regulatory proteins. Among these, heat shock protein-90 kD (HSP90), which binds both GR and MR to the C-terminal domain, heat shock protein-70 kD (HSP70), p59 immunophilin, Fkbp51 and Fkbp52 and the small p23 phosphoprotein maintain correct protein folding of the receptor [102,112,113]. The GR, which belongs to the nuclear receptor transcription factor family, is composed of different conserved structural domains [114]. These include an N-terminal variable region (NTD) required for ligand-independent gene transactivation, which contains a transactivation domain named activation function 1 (AF1). The latter is responsible for the transcriptional activation and is involved in the association with coregulators and the basal transcription machinery. A central DNA-binding domain composed of two zinc fingers has been shown to be crucial both for GR homodimerization and DNA-binding specificity. This is followed by an adjacent flexible hinge region allowing proper DNA binding, dimerization, and nuclear translocation of the receptor [115]. Finally, the C-terminal region (LBD) contains the ligand binding domain and a secondary transactivation domain (AF2), regulated by hormone binding, which is essential for dimerization, interaction with cochaperones, coregulators, and other transcription factors [116,117]. The LBD also comprises a dimer interface which is fundamental for GR function and the binding of the heat shock protein 90 (Hsp 90) [118]. Both DBD and LBD include nuclear localization signals, which are required for GR nuclear translocation. Finally, DBD also incorporate the nuclear export signal sequence (NES) which targets it for export from the nucleus to the cytoplasm via the nuclear pore complex [113] (Figure 1).

Although the NTD is conserved, literature reviews and sequence alignments of human, monkey, rat, and mouse GRs have revealed that there are another eight conserved AUG start codons in the exon 2 (Figure 1). In humans, these were shown to produce various GR isoforms with progressively shorter N-terminal transactivation domains [113]. These are formed due to the presence of alternative Kozak translation initiation sequences which can cause either ribosomal shunting or ribosomal leaky scanning mechanisms. This allows the generation of different GR subtypes with truncated N-termini [119,120,121,122,123], which are likely to be fully active. This is consistent with data from zebrafish, where a GR mutant line (*gr^sh551^*), characterized by a 1 bp deletion in the first coding exon (exon 2, Q48fsX3), proved not to have any detectable phenotype [124]. This was confirmed further by ISH analysis, which showed that both *gr^sh551^* mutants and wildtypes displayed an identical downregulated *pomca* expression after synthetic GC (Betamethasone 17,21-dipropionate) administration.

Synthetic GR agonists are supposed to trigger a potent GC response, which in turn elicits the GC-GR mediated negative feedback loop, aimed to shut down their own biosynthesis. As previously mentioned, this mainly occurs at the level of the pituitary gland via downregulation of *pomca* [2,106,111]. For this reason, if GR is not functional, the GC-GR negative feedback loop cannot occur and *pomca* expression should not be downregulated, as occurs in *gr^sh551^* mutants. Indeed, in *gr^sh551^* mutants the feedback occurs normally.

In addition to alternative starts, alternative splicing at exon 9 is responsible for generating two different GR splice variants, namely GRα (777 aa) and GRβ (742 aa) [25,125]. These two receptor isoforms share an identical amino acid sequence between 1–727 aa and then diverge. In particular, the human (h) GRα c C-terminal region contains 50 distinct amino acid residues that form two alpha-helical structures that play a key ligand binding role. In contrast, the hGRβ C-terminal is characterized by a shortened 15 non-homologous, specific amino acid sequence that prevents GC binding [20,126,127]. Hence, hGRβ does not bind traditional glucocorticoid agonists and lacks transactivational activity on GRE-containing promoters, whereas hGRα is the canonical GR isoform. Nevertheless, hGRβ is constitutively present in the nucleus where it has been shown to act as a dominant-negative inhibitor of hGRα’s transactivational properties [127,128,129]. The mechanism behind this inhibition is still uncertain, but several studies have suggested that competition between both hGR isoforms for transcriptional coactivator proteins and/or the formation of inactive GRα-GRβ heterodimers might be responsible for that [126,130,131,132].

Moreover, the function of hGRβ extends beyond antagonism of the hGRα isoform [133]; for instance, binding to the glucocorticoid antagonist mifepristone (RU486) has been also reported [134,135]. In addition, it has been shown that increased hGRβ expression is correlated both with the development of immune-related diseases (e.g., ulcerative colitis, leukemia and severe asthma) [136,137,138] and with glucocorticoid resistance in patients affected by these diseases [139,140,141]. Interestingly, previous studies have established the occurrence of a GR β-isoform in zebrafish larvae, which similarly to the hGRβ, confirmed the lack of a role in transcriptional regulation and a dominant-negative inhibitor activity on zGRα [19,20,142,143]. In this respect, zebrafish have been shown to be a reliable and useful model system both for GC resistance and glucocorticoid receptor research [2,85].

### 2.3. GCs Mechanisms of Action

The conventional view of GC mechanism of action has been recently revised and described as a more complicated multiprotein-regulated process. In this regard, it has been shown that in both humans and zebrafish, upon cortisol binding, GR undergoes a conformational change that involves an FKBP51-FKBP52 exchange. The latter triggers the translocation of the GC-GR active complex into the nucleus. FKBP51 is a cochaperone protein that binds HSP90 and decreases the affinity of GR for cortisol. For this reason, FKBP51 has been considered an inhibitor of GR transcriptional activity and its overexpression has been linked to GC resistance in autoimmune diseases [144,145,146]. After ligand binding, FKBP51 is replaced by FKBP52, which in turn recruits dynein to support translocation of the GC/GR complex to the nucleus [147,148]. This structural modification exposes the two GR nuclear localisation signals, allowing the hormone-activated GR to dimerize with another GC-GR molecule and to migrate into the nucleus via nuclear pores [149,150]. Interestingly, this transcription factor complex can also act nongenomically (in the cytoplasm), where it may interact via direct protein-protein interactions with other transcriptional regulators and/or kinases (e.g., basal transcription machinery (BTM); phosphoinositide 3-kinase (PI3K); signal transducer and activator of transcription (STAT)) [63,151,152,153] (Figure 2).

Inside the cell nucleus, GC-GR complexes can directly bind to specific GREs, as tetramers, to upregulate (transactivation) or downregulate (transrepression) the transcription of target genes. Generally, the preferred GRE motif (AGAACAnnnTGTTCT in humans, and GGAACAnnnTGTTCT in zebrafish) is an imperfect palindromic consensus sequence that consists of two 6 bp half sites. The three-nucleotide spacing in-between the two half sites is essential for the GR to tetramerize on this sequence. Previous genome-wide studies have shown that the same GRE can mediate both the GC-dependent induction of many genes (positive GRE) and the repression of others (negative GRE) [113,154]. Interestingly, the presence of specific inverted repeats negative GREs (IR nGRE), unrelated to simple GREs has also been reported both in mice and in humans. These DNA binding sequences are palindromic sequences consisting of two inverted repeated (IR) motifs separated by 1 bp. In particular, they bind GC-GR complexes to promote the assembly of cis-acting GR-SMRT/NCoR repressing complexes [155,156] (Figure 2).

In summary, these findings indicate that the broadly different GC effects on various tissues can be partially ascribed to cell type-specific differences in the chromatin landscape that affects the accessibility of specific GREs for GR binding [157,158]. Furthermore, the GC concentration at which the GR binds to GREs depends on the cell type and chromosomal context. Another important feature of the GC-GR complex that makes its effects even more versatile is that it can tune gene expression in different ways: by binding directly to DNA, by tethering itself to other transcription factors bound to DNA, or via direct binding to DNA and with neighbouring DNA-bound transcriptional regulators (composite manner, Figure 2) [113,159,160].

### 2.4. The Mineralocorticoid Receptor: Structure and Functions

As previously mentioned, cortisol can bind not only to GR, but also to MR. Both are members of the steroid receptor superfamily (corticosteroid receptors) of ligand-activated transcription factors that enhance or repress the transcription of target genes, as well as promote rapid nongenomic/extra-nuclear events via several cell signalling pathways [47]. The mineralocorticoid receptor is a 984-amino acid cytoplasmic protein that is characterized by three different domains: an N-terminal transcriptional regulator domain, a DNA-binding domain, and a ligand-binding domain responsible for the selectivity of hormone binding. Analogously to GR, in its unliganded state MR is associated with a number of chaperone proteins (HSP90, HSP70, FKBP51 and p23) that play a crucial role in trafficking and maintaining MR in a suitable conformation for ligand binding [161].

MR has a ten-fold higher affinity for cortisol than GR and is preferably activated under basal conditions, implying distinct roles for each receptor in the regulation of HPA axis activity [162,163] It has also been observed that cortisol, even at lower concentrations than those required to activate the GR, binds to MR and might enhance the activity of several kinases (i.e., protein kinase C (PKC), cyclic adenosine 3′,5′-monophosphate (cAMP), and phosphoinositide 3-kinases (PI3K)) involved in different signal transduction cascades [4,47,164]. On the other hand, GR is primarily activated as a result of stress, or at the diurnal peak when circulating cortisol levels are also peaking [162,165]. Cortisol also exerts broad effects on mood and behaviour via MRs and GRs that are expressed in different regions of the brain [166,167]. In particular, GR is widely expressed throughout the brain, primarily in the PVN (stress-regulating centre) and in the prefrontal cortex-hippocampal-amygdala circuitry (cognitive, emotional regulation and memory consolidation centre). Vice versa, MR is predominantly expressed in the hippocampus, amygdala, and the lateral septum (emotionality, social behaviour and feeding process hub) [168].

In mammals, the mineralocorticoid system is essential to regulate potassium and fluid homeostasis upon aldosterone activation of MR. Even though cortisol is a high-affinity ligand for MR, this steroid is deactivated in specific mineralocorticoid responsive tissues, such as the kidney, by the previously mentioned 11b-HSD-2 enzymatic activity. This allows aldosterone, a second corticosteroid present in mammals, to bind to this receptor. Surprisingly, teleosts do not synthesize aldosterone, and cortisol has been shown to mediate stress axis regulation, as well as the majority of the changes in iono-regulatory and osmo-regulatory functions, via GR and MR signalling [1,4].

Interestingly, previous work performed both in rats and in teleosts showed that while MR is involved both in basal and onset of stress-induced HPA/I axis activity, GR mainly controls its termination [4,169]. However, a recent zebrafish study highlighted that rapid locomotor responses to quick changes in light illumination or water salinity (environmental stressors) require GC-GR mediated HPI axis signalling, but not MR [170]. Finally, our recently published research suggested that not only GR, but also MR signalling, is involved in the GC-negative feedback regulation (HPI axis termination) and plays a key role in assuring a proper HIF response in teleosts [171]. In view of this, furthering the precise role of MR and mineralocorticoid modes of action in vivo, particularly in relation to in the HIF signalling pathway, is warranted.

### 2.5. The Role of Glucocorticoids in Inflammation

Glucocorticoids suppress most of the events early in the inflammatory response and subsequently facilitate inflammation resolution. They suppress both vasodilation and the enhanced vascular permeability that occurs as a consequence of an inflammatory challenge, and inhibit leukocyte migration from the inflamed region [10,172,173]. In addition, they tune both distribution and trafficking of leukocytes, promote death/survival in some cells and may affect cellular differentiation programmes [173,174,175,176,177,178]. It is well established that many of the anti-inflammatory and immunosuppressive glucocorticoid actions are referable either indirectly or directly to GC-GR mediated transcriptional regulation of numerous genes expressed in leukocytes [173]. On the other hand, even though MR expression has been reported in immune cells [179,180,181], its anti-inflammatory role has been considered negligible so far [181,182], but rather surprisingly, MR-dependent proinflammatory effects have also been noted [183,184]. It is presently unclear whether these effects are mediated by the glucocorticoid cortisol or the mineralocorticoid aldosterone [185].

Glucocorticoids, through GR, can modulate gene expression in different ways [112,117]. Among these, transactivation is the mechanism by which GC predominantly induce the transcription of numerous anti-inflammatory genes, such as GILZ, MKP-1 and IĸBα. This occurs through direct binding of single or multiple GC-GRs to palindromic glucocorticoid response elements (GREs) [186,187,188,189]. Importantly, this mode of action has also been shown to be responsible for several undesirable metabolic side effects linked to chronic GC treatment [148]. On the other hand, transrepression is the mechanism by which GC downregulate the transcription of inflammatory genes and requires direct protein-protein interaction of GR to other transcription factors. This mode of action is generally accepted to convey the beneficial GC anti-inflammatory effects, which are mainly implicated in rapid cellular responses [40,148,190,191,192]. In particular, transrepression is known to mainly occur via direct binding between the monomeric GC-GR complex and transcription factors (e.g., AP-1, NF-kB, c-Jun, and c-Fos) activated by cytokines and other pro-inflammatory stimuli, which synergistically coordinate the expression of several proinflammatory genes [27,109,160,193,194,195,196]. Of note, most of these genes are commonly overexpressed during chronic nonresolving inflammatory states. Interestingly, transrepression is not restricted to these transcription factors but also includes others such as CREB, STAT, and T-bet [160,197]. As a result, the mutual antagonism between transcription factors frequently impairs their transcriptional properties and prevents them from binding to their corresponding DNA response elements.

As a consequence of the above considerations, optimal GC analogs should be characterized by a high inhibitory activity against inflammatory mediators, coupled with a low transactivation activity, in order to induce minimal side effects. Interestingly, different steroidal and nonsteroidal ligands have been reported to have this dual function (e.g., RU-24858 and ZK-216348) [148,191,198,199,200]. These compounds have been reported to suppress key inflammatory and immune transcription factor activity in vivo [198,199,200,201,202]. However, as stated before, since GCs can trigger gene expression via multiple routes, unexpected secondary side effects might occur. For this reason, further research is also warranted to elucidate the implications of the nongenomic GC-mediated activity both in the immune and inflammatory scenario.

Importantly, although the GC-GR complex is known for its anti-inflammatory effects, the picture is more complex. Indeed, contrary to expectations, GR loss of function was reported by Facchinello and coworkers to prevent the transcriptional activity linked to the inflammatory immune response (i.e., of cytokines Il6, Il1β, Il8 and Mmp-13) [8], corroborating the hypothesis of a GC-GR mediated dual action on the immune system [35,203]. However, it is clear that further research is warranted to better elucidate this aspect. In addition, GR was shown to synergistically induce proinflammatory genes by acting on other signalling pathways [204,205,206,207]. Finally, studies also demonstrated that GCs increase the transcription of numerous anti-inflammatory molecules such as interleukin-10 (IL-10), interleukin-1 receptor antagonist (IL-1RA), secretory leukocyte inhibitory protein and neutral endopeptidase [41,208].

Previous research also revealed that alterations in chromatin structure are important for modulating the outcome of GC activity. Indeed, GR can interact differently with histone acetyltransferases (HATs), with histone deacetylases (HDACs) and also kinases (i.e., MSK1, PKA and JNK) [160]. These may, in turn, modulate the chromatin environment by modifying chromatin accessibility and further tuning inflammatory and immune gene expression [160]. Furthermore, since chromatin accessibility can predetermine GR binding patterns and is crucial for cell-specific outcomes, it can provide novel molecular basis for tissue selectivity [209,210]. In addition, another study showed that GR may directly inhibit CREB binding protein (CBP)-associated HAT activity and may recruit HDAC2 to the p65-CBP HAT complex [211]. This novel glucocorticoid repression mechanism suggests that histone acetylation inhibition represents an additional level of control of inflammatory gene expression. Consequently, this further indicates that pharmacologically manipulating specific histone acetylation status could be an alternative approach for treating inflammatory diseases.

## 3. The HIF Signalling Pathway

The progression of the cell cycle is an energy-requiring process that demands a refined metabolic regulation to occur. Indeed, it is well known that cells must overtake an energy restriction checkpoint during G1 phase, in order to progress through the cell cycle. In this regard, as practically all metazoan organisms require oxygen for metabolically converting nutrients into energy, O_2_ represents a vital signalling molecule directing cellular functioning and survival. Consequently, assuring oxygen homeostasis is a critical task that must be precisely managed by cells in order to perform correctly and survive in a hostile environment [212].

In vertebrate cells, this is primarily carried out by Hypoxia-Inducible Factors (HIFs), which are a family of transcription factors that react both to environmental oxygen and cellular energy alterations (e.g., hypoxia) [213]. HIFs are obligate heterodimers consisting of three main α-subunits (HIF1α, HIF2α and HIF3α) and two constitutively stable aryl hydrocarbon receptor nuclear translocators or β-subunits (ARNT1 and ARNT2). Both HIF alpha and beta subunits are expressed in the cytoplasm and are basic helix-loop-helix (bHLH)/Per-Arnt-Sim (PAS) transcription factors containing transactivation domains (TADs) [214]. In particular, the initial bHLH domain, required for DNA binding, is followed by a Per-Arnt-Sim (PAS) domain that acts as a molecular sensor. The latter is followed by an oxygen-dependent degradation domain (ODD), which is targeted and hydroxylated by the VHL-E3-ubiquitin ligase complex, and by an N- and C-terminal transactivation domains (TAD-N and TAD-C). The latter is present in all the HIF-α subunits except for HIF-3α, which only has the TAD-N activation domain. On the other hand, HIF-β/ARNT being not targeted and degraded by the pVHL-E3 ligase complex lacks both the ODD and the TAD-N domains, and has a constitutively active N-terminal nuclear localization signal (NLS) [215,216] (Figure 3A).

As previously mentioned, HIF-α subunits are characterized by a very fast turnover that is post-translationally regulated by the PHD3-VHL-E3-ubiquitin ligase protein degradation complex. Under normoxic conditions (normal oxygen levels), a set of prolyl hydroxylases (PHD1, 2 and 3) directly exploit the available molecular oxygen to hydroxylate two prolyl residues (Pro402, Pro564) within the oxygen-dependent degradation domain (ODDD) of the HIF-α subunits (Figure 3A). Then, the hydroxylated HIF-α isoforms are recognized and targeted by the Von Hippel Lindau protein (pVHL), which acts as the substrate recognition component of the E3-ubiquitin ligase complex. The latter is characterized by a multiprotein complex which consists of Elongin B, Elongin C, Ringbox 1 and Cullin 2 [217]. Once this complex ubiquitinates the HIF-α subunits, it directs them towards proteasomal degradation to avoid an aberrant stabilization and activation of the HIF pathway when it is not necessary (Figure 3B, normoxia). In addition, HIF-α may be hydroxylated by factor-inhibiting HIF (FIH) on asparagine 803 (N803), which prevents the recruitment of the transcriptional coactivator p300/CREB-binding protein (p300/CBP) and reduces the effectiveness of HIF transcriptional activation [218].

Another level of complexity in the regulation of HIF response is represented by the fact that both FIH and PHDs are dioxygenases requiring O_2_, ferrous iron (Fe^2+^), 2-oxoglutarate and ascorbate, as cosubstrates, to hydroxylate HIF-α [219]. The ferrous iron is an essential cofactor for the enzyme to be assembled into its active conformation, as prolyl hydroxylases contain Fe^2+^ in their hydrophobic active centre. Moreover, the O_2_-binding event requires ascorbate to maintain this iron molecule into its ferrous state, whereas the transferring of one oxygen atom to 2-oxoglutarate (2-OG) is required to hydroxylate HIF-α subunit. This reaction yields both succinate and carbon dioxide as reaction products, making this catalytic process irreversible. During a complete reaction, Fe^2+^ is transiently oxidized to Fe^4+^ and then reduced to the Fe^2+^ state. Interestingly, when α-ketoglutarate is converted into succinate without hydroxylation of a peptide substrate, Fe^2+^ is oxidized to Fe^3+^. In this process, ascorbate is also necessary to reduce Fe^3+^ back to Fe^2+^ to allow the enzyme to be recycled [220]. Finally, mitochondrial reactive oxygen species (ROS), may also affect hydroxylase activity by post-translationally modifying these enzymes by oxidizing their cysteine residues, and/or by attacking ferrous iron (Fe^2+^) [221,222].

By contrast, the presence of reduced O_2_ levels impairs both PHD and FIH enzymatic activity and leads to HIF-α stabilization. Therefore, pVHL is no longer able to recognize and target HIF-α to proteasomal degradation when the latter is not hydroxylated. This allows HIF-α and β subunit heterodimer formation, followed by ARNT-mediated translocation in the nucleus. Here, p300/CBP may interact with the HIF-αβ transcription complex to further activate the hypoxic response. This implies the upregulation of target genes that are involved in decreasing oxygen consumption and increasing oxygen and nutrient delivery [31]. This occurs via direct recognition and binding of HIF to hypoxia-response elements (HREs). The latter are characterized by the presence of a consensus sequence G/ACGTG located within the promoter regions of target genes, such as phosphofructokinase, adrenomedullin, erythropoietin and vascular endothelial growth factor [73,74,78] (Figure 3B, hypoxia).

### The Role of Hypoxia in Inflammation

Hypoxia and inflammation share an intimate relationship. Both in cells and tissues the hypoxic response plays a vital role in the metabolic changes that control cellular adaptation to low oxygen availability. In a hypoxic scenario, HIF exerts a proinflammatory role by stimulating multiple aspects of the host immune system, from enhancing the phagocyte microbicidal capacity, to driving T-cell differentiation and cytotoxic activity [223,224]. The HIF-mediated activation of the inflammatory response is a complex process, which is characterized by the simultaneous activation of both pathways in several pathological circumstances such as chronic inflammation, wound healing and solid tumours [225]. Hypoxia is able to activate monocytes, macrophages and dendritic cells by tuning their gene expression and cytokine secretion [226,227,228,229]. In addition, it triggers NF-κβ stabilisation, which acts as a master regulator of the inflammatory and anti-apoptotic response. This is achieved, as for HIF-α, via the oxygen dependent inhibition of prolyl hydroxylase activity. Then, this triggers the decrease in IκB kinase beta (IKKβ) hydroxylation, which leads to the activation of NF-κβ [87,230,231,232]. In turn, in response to hypoxia, LPS, or bacterial infection, NF-κB has been demonstrated to directly increase HIF-1α transcription [233,234,235]. This allows further confirmation of the extensive crosstalk between these two major molecular players involved in inflammation and hypoxia [236]. This interaction has been well studied in a zebrafish model of wound healing, where HIF-1α pathway activation was observed to delay neutrophil resolution [82]. This is believed to occur as a consequence of HIF activation inside the neutrophils themselves, and seems to be related to an augmented neutrophil apoptosis rate coupled with decreased trafficking away from the comorbid site [237].

Interestingly, several studies have also reported the presence of an interplay between hypoxia and glucocorticoid-dependent signalling pathways. Indeed, glucocorticoids have been observed via in vitro studies both to enhance and inhibit HIF pathway activation [77,78,79,80,81]. Analogously, hypoxia has been shown to attenuate the glucocorticoid anti-inflammatory response and to elicit corticosteroid insensitive inflammation [238,239,240].

In this regard, it is important to note that despite hypoxia being considered pro-inflammatory, HIF can also positively interact with pathways with apparently opposite effects. For instance, it was shown that HIF and GC signalling converge at the level of the promoter region of inflammatory factors to reciprocally tune their expression in a T-lymphocyte cell line model [241]. In relation to inflammation regulation, particular attention should be also placed on GILZ [242], which is an important mediator of the GC anti-inflammatory and immune-suppressive activity. Indeed, GILZ overexpression in T-cells has been demonstrated to suppress NF-κB activation by binding to its p65 subunit and preventing its nuclear translocation [243,244]. Moreover, it has been reported that hypoxia can strongly upregulate GILZ expression in rat macrophages, and that pretreatment with synthetic GC (Dexamethasone) amplifies these effects [242]. Additionally, in vitro studies revealed that under hypoxic conditions GILZ inhibition led to increased transcription and protein secretion both of proinflammatory mediators IL-1 and IL-6 and abolished the GC inhibitory effect on their expression. These findings indicate that GILZ plays a key role in tuning adaptive responses to hypoxic conditions by inhibiting pro-inflammatory responses and by mediating the GC anti-inflammatory responses [242].

Macrophages, which are broadly classified as M1 (pro-inflammatory), or M2 (anti-inflammatory) are a class of immune cells, residing in all tissues, that play a crucial role during inflammation [245]. In addition to their protective immunological function they also stimulate growth factors and pro-angiogenic cytokine expression such as basic fibroblast growth factor (FGF-2) and vascular endothelial growth factor (VEGF-A), thereby triggering angiogenesis [246]. Interestingly, under hypoxic conditions, HIF can also promote their polarization towards the M2 phenotype in order to modify the inflammatory microenvironment by decreasing the release of proinflammatory cytokines [247].

Conversely, GCs, being angiostatic, are often exploited to treat angiogenesis-related diseases, including solid tumours. In particular, GCs control angiogenesis by inhibiting proliferation, migration and sprouting of endothelial cells, and by decreasing both cytokines and pro-angiogenic factor expression [248]. Consequently, since both hypoxia and GC are specifically involved in inflammation, where they tune angiogenesis and affect macrophage function (i.e., by upregulating GILZ expression), understanding how precisely this interplay occurs in vivo, may have a wide physiological significance in health and disease, and may help researchers to develop more effective anti-inflammatory drugs in the future.

## 4. HIF-GC-Crosstalk: Previous Insights

Modulating the HIF pathway has the potential to be clinically exploited as therapeutic treatment for a variety of pathological conditions which include stroke, ischemia, spinal cord injury, inflammation, cancer, wounding, chronic anaemia and bone regeneration [99,224,249,250,251,252]. To this end, an unbiased chemical screen performed in our laboratory on zebrafish larvae discovered that HIF associated transcriptional responses are potently activated by GCs, particularly in the zebrafish liver [253].

Moreover, by translating these observations to human tissues, it has been possible to show that GCs promote HIF stabilization, without the need of the GR DNA binding domain (non-genomic action), in primary human hepatocytes and intact liver slices. In this regard, since c-src inhibitor PP2 treatment was able to rescue this effect, this suggested a role for GCs in promoting c-src–mediated proteasomal degradation of pVHL followed by stabilization of HIF-α subunit [253]. According to these data, since the liver is an important regulator of blood glucose levels, and both GC and HIF promote gluconeogenesis and glycogen storage in the liver, the crosstalk between these may contribute to GC effects on glucose metabolism and may have a wider physiological significance in health and disease than previously expected.

Indeed, both GCs and hypoxic transcriptional responses are mutually involved in assuring tissue homeostasis by controlling cellular responses to various forms of stress and inflammation, especially affecting glucose metabolism.

Synthetic GCs have been widely used for years as anti-inflammatory drugs for treating pathological conditions in which hypoxia plays a role in disease progression, such as rheumatoid arthritis and chronic obstructive pulmonary disease [31,32,35]. In humans, the adaptive GC release in response to atmospheric hypoxia has been shown to be linked to acclimatisation to high altitude, and the prophylactic treatment with GC has been broadly exploited to mitigate the related mountain sickness [254]. Additionally, GC administration protects different organs from ischemic injury, as was observed especially against experimental cerebral and hepatic ischemic/reperfusion injury [255,256,257,258].

The presence of an interplay between hypoxia and GC-dependent signalling pathways has been previously reported in different in vitro studies [78,79,80,259]. However, these studies reported conflicting results on the crosstalk between GC action and hypoxia, where the latter limits GR-mediated transactivation both in pulmonary endothelial and hepatic epithelial cells [78,80]. The first data about the interaction between HIF and GR were presented by Kodama et al., 2003 [79]. By exploiting an artificial approach using Gal4-fusion reporter assays, they found that the ligand-dependent activation of GR increases hypoxia-dependent gene expression and hypoxia response element (HRE) activity in HeLa cells.

Moreover, using dexamethasone-treated COS7 cells co-transfected with expression plasmids for either GR or GAL4-LBD and GFP-HIF-1α and exposed to hypoxic conditions, they showed colocalization of the GR and HIF-1α in the nucleus. For this reason, Kodama and colleagues postulated the presence of a direct protein-protein interaction between the GR LBD and HIF-1α as the main mechanism for GC-dependent enhancement of the HIF pathway but failed to demonstrate it via GST pulldown assays.

Leonard et al., 2005 [78], subsequently confirmed via microarray analysis that GR is upregulated (from 7-fold to 12-fold) by hypoxia over time (0–36 h) in human renal proximal tubular epithelial cells. Furthermore, using a cell-based GRE luciferase reporter system, they showed that hypoxic exposure can potentiate dexamethasone-stimulated GRE promoter-reporter activity.

Using ACTH secreting mouse pituitary tumor AtT-20 cells, Zhang et al., 2015 [259] demonstrated that GR expression levels were enhanced by HIF-1α under hypoxic conditions. However, dexamethasone treatment was able to cause the downregulation of GR expression in a HIF-1α dependent way. Finally, even if this was confirmed by transfecting AtT-20 cells with HIF-1α siRNA and culturing them under normoxia or hypoxic conditions, the involved underlying mechanism remains unclear. Unfortunately, the impact on downstream GR/GC pathway activity was not assessed.

By contrast, a dexamethasone-related inhibition of HIF-1α target gene expression in hypoxic HEPG2 cells was revealed by Wagner et al., 2008 [80]. In particular, via Western blot analysis they showed that dexamethasone reduces nuclear HIF-1α protein, as the HIF-1α amount was higher in cytosolic cell extracts than in the nuclear extracts upon DEX treatment. This cytoplasmic retention of HIF-1α suggested a blockage of nuclear import, via a still unknown mechanism, which resulted in a reduced HIF target gene expression. In addition, by exploiting a luciferase assay the author revealed that dexamethasone attenuates HIF-1 activity not only in a GR- dependent way but this effect depends on the presence of functional HREs. Importantly, Wagner et al. attributed these contradicting results (compared to Kodama’s) to the fact that they conducted all their cell-culture experiments in the presence of fetal bovine serum (FBS), avoiding not only growth and cell-cycle arrest, but also the synchronization of the cells. This might be an important issue, as many cellular processes depends on the cell cycle phase.

In the following years, Gaber et al., 2011 [260] investigated the interaction between macrophage migration inhibitory factor (MIF), HIF and the effect of GCs in human primary nontumor CD4+ Th cells and Jurkat T cells. In contrast to the previous observations, this study showed the presence of a clear dexamethasone-dose dependent inhibition of HIF-1α protein expression, which resulted in decreased HIF-1 target gene expression. Therefore, as an alternative to the Wagner et al. 2008 hypothesis, they proposed a model based either on rapid DEX-mediated induction of HIF-1α inhibitors (e.g., PHD1–3, FIH, and pVHL) or fast DEX-mediated suppression of hypoxia-induced signalling. Cumulatively, these contradicting in vitro data reflect the diversity present among the cell types that were used in these studies and the different mechanisms by which these were conducted. Methods that are less cell line and cell culture-dependent would be helpful to understand the interaction better, and in vivo analysis would be valuable, in this respect.

## 5. HIF-GC-Crosstalk: HIF as Negative Regulator of GC Biosynthesis and Responsiveness

Zebrafish have been demonstrated to be informative organisms in which genetic mutations and chemical modulators of HIF and GC signalling pathways can be easily combined, not only in physiological, but also in pathophysiological conditions. Using such approaches [89,91,261,262], we demonstrated that the upregulation of HIF signalling repressed both GR responsiveness and cortisol levels, whereas GCs enhanced HIF activity. The latter effect was mediated both by the GR and MR receptors, and we speculated that GC can promote HIF-1 signalling via multiple routes [171].

With respect to the effect of HIF signalling on GC responsiveness, it was particularly interesting to note that a strong activation of HIF signalling can blunt GR transcriptional regulation, whereas reduction of HIF signalling via *arnt1* loss of function is able to derepress it. Interestingly, HIF activity leads to a repression of GC biosynthetic genes, which is unusual since HIF is generally considered to be a transcriptional activator. However, the precise molecular mechanism behind this the strong negative HIF-mediated action remains to be resolved.

As stated before, GCs control a plethora of physiological processes, act on almost all the tissues and organs in the body and have a strong anti-inflammatory and immunosuppressive actions. For this reason, their production must be finely controlled by the HPA/I axis [113]. If the latter is disrupted by factors such as chronic stress, disease progression and prolonged exogenous GC treatment, a tissue-specific decrease in GRα functional pool coupled with an uncontrolled GC biosynthesis may result in the development of acquired GC resistance. This results in both excessive inflammation and HPA axis hyperactivity, which are known to contribute to the progression of numerous psychological and pathological conditions (e.g., cardiovascular disease, depression, schizophrenia, diabetes, and cancer, among others) [263,264].

Cumulatively, the presence of high cortisol levels coupled to a significant decrease in GC sensitivity are seen as a hallmark of glucocorticoid resistance (e.g., in the gr mutant fish). Therefore, it is rare that a GC-resistant condition coincides with low cortisol levels. In addition, many studies have shown that if GC levels drop, *pomca* is upregulated to compensate for the reduced GC levels and feedback [2,87]. However, in the presence of upregulated HIF pathway (in *vhl* mutant fish), an unusual situation is seen where *pomca* is not upregulated, although GR-responsiveness is strongly attenuated, and GC levels are very low. Consequently, we speculate that HIF signalling can act as a second upstream controller of the GC-mediated stress response (in addition to cortisol levels themselves) (Figure 4).

Indeed, since previous work in our laboratory highlighted that GCs may also act as HIF activators [76,253] we inferred that HIF may, in turn, control cortisol levels by acting on *pomca* expression. This would allow HIF signalling not only to manage its own activity, but also to assure both stress resolution and homeostasis. The reason for hypothesising that HIF signalling would counteract anti-inflammatory GC activity resides in the fact that the simultaneous expression of both the upregulated HIF and GC pathways would be detrimental to homeostasis. This is because HIF is a master regulator of cellular pro-inflammatory responses to hypoxia, whereas GCs have a potent anti-inflammatory and immune suppressive activity [171]. In this regard, it is interesting to note that our proposed model effectively applies to the immune response context. Indeed, during inflammation, we speculate that the crosstalk between GC and HIF may influence the expression of downstream inflammatory effectors (e.g., GILZ and NF-κB) to eventually assure homeostasis (Figure 5).

This hypothesis is in accordance with a previous report showing that hypoxia exposure resulted in the downregulation of steroidogenic genes (*StAR*, *cyp19b, cyp19a, cyp11c1, hsd17b2 and hmgcr*) in 72 hpf larvae, whilst zHIF-α loss of function stimulated the upregulation specifically of *StAR, cyp11b2 and cyp17a1* [265]. Importantly, the fact that cortisol levels were reduced in *vhl-/-* and upregulated in *arnt1-/-* is consistent with our assumption. More speculatively, the fact that in teleosts GC biosynthesis is finely regulated by hypothalamus-pituitary gland activity in a circadian way, and multiple studies have shown an evolutionary connection between HIF signalling and circadian rhythms [266,267,268,269], suggest that the interaction between GC and HIF may be tight and ancient.

Importantly, recent work from Watts et al., 2021 confirmed the role of HIF1α as a direct regulator of steroidogenesis in the adrenal gland. Indeed, mice deficient in HIF1α in adrenocortical cells exhibited both increased levels of steroidogenic enzymes, such as Star, Cyp11a1, Cyp21a1 and Cyp11b1, and an enhancement in circulatory steroid levels. These changes also resulted in cytokine alterations and modifications of the profile of circulatory mature hematopoietic cells. However, additional mechanisms are possible and indeed likely, one example being via miRNA103/107 [270]. Vice versa, HIF1α overexpression induced the opposite phenotype characterized by strongly downregulated steroid production as a result of impaired transcription of steroidogenic enzymes [271]. Therefore, the data derived from zebrafish research are in accordance with mice studies. As a consequence of the above considerations, the HIF-mediated *pomca* negative regulation seems to be a logic homeostatic response. Moreover, with respect to the stimulating effects of GCs on HIF signalling, it is important to point out that in vivo genetic analysis showed that functional GR is an essential prerequisite for high HIF signalling levels. Unfortunately, a molecular explanation for this is still elusive, and research towards this direction (e.g., ChIP seq, pull-down assay) is warranted.

Finally, the importance of GCs in the HIF pathway was further analysed by studying MR contribution to HIF signalling itself. Our data showed that in addition to the glucocorticoid receptor, the mineralocorticoid receptor acts partially redundantly to allow HIF signalling in the zebrafish. This hypothesis is consistent with work from Faught and Vijayan [4] showing that both Gr and Mr signalling are involved in the regulation of the GC negative feedback. It is also consistent with the finding from Rybnikova et al., that both GR and MR are involved in brain hypoxic tolerance induction [272]. If these results hold up in mammals, this suggests GR and MR inhibitors may be usable to reduce excessive HIF activity in certain situations; for instance, in *vhl* deficient tumours.

In addition, dexamethasone (GC agonist) has been recently shown to decrease short-term mortality and to reduce the need for mechanical ventilation in hospitalised patients with COVID-19 [273,274]. Since the latter show low blood oxygen saturation, and HIF is very likely to be overactivated in them, could it be that they have either too low GC levels or GR responsiveness? Since synthetic GC administration rescues these patients with upregulated HIF, is there a functional link between these two facts? Is this related to what we have seen in the zebrafish?

Future work that could be brought to any potential drug development would include additional preclinical testing both in vitro and in vivo, aiming to better elucidate how HIF-GC interaction take places at molecular level. Cumulatively, understanding the molecular mechanisms behind the HIF-GR crosstalk and their target gene activation is of undeniable importance and deserves significant research investment in the near future.

## 6. Conclusions and Future Directions

The development of novel therapeutic targets and of innovative GCs with a better benefits-risk ratio is of indisputable relevance to effectively treating immune and inflammatory responses. In this regard, the zebrafish has been shown to be an effective model organism that can be used for investigating not only the molecular mechanism behind glucocorticoid receptor modes of action, but also in drug discovery studies. To this end, it is important to define efficient readouts for both desired and undesired effects of GC.

In the last decades few in vitro studies have highlighted the potential for crosstalk between the hypoxia-inducible factor and glucocorticoid transcriptional responses. However, how this interplay takes place precisely in vivo has only been recently proposed, and use of the zebrafish as a model organism has shown that HIF can repress not only cortisol biosynthesis, but also GR responsiveness to synthetic GC. Conversely, research has uncovered the importance of the GC pathway in driving HIF signalling and highlighted a novel mineralocorticoid receptor contribution to the HIF-GC crosstalk. In addition, the interaction between HIF-GC and NF-kB signalling might be of high relevance. Indeed, investigating the putative mechanisms behind adaptation to severe hypoxia might help to unravel the hypoxia-related causes and effects in acute inflammatory disease. Finally understanding how MR precisely impacts on HIF signalling might be extremely important to provide a potential additional avenue to downregulate the HIF pathway in vivo, as it has been proven difficult to do so in the tumour microenvironment. Overall, new insights into the interaction between HIF and GR, both in vitro and in vivo, are promising directions for future research and might inspire new therapeutic approaches.

## Figures and Tables

**Figure 1 cells-10-03441-f001:**
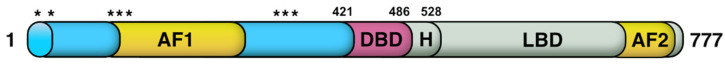
Glucocorticoid receptor domain structure and translational isoforms. The N-terminal domain (NTD), which is required for ligand-independent gene transactivation, includes a transcriptional activation function region (AF1). The latter, which interacts with coregulators and with the basal transcriptional machinery, is the main posttranslational modifications site. The LBD, which is made up of 12 α-helices and 4 β-sheets, forms a hydrophobic pocket needed for GC binding and includes an AF2 domain. The latter allows interaction with coregulators in a ligand-dependent way. Finally, two nuclear localization signals, named NL1 and NL2, are localized in the DBD-hinge region junction and within the LBD, respectively. Asterisks indicate the location of the starting amino acid (aa position: 1, 27, 86, 90, 98, 316, 331, 336) of the eight different GRα translational isoforms, which are characterised by progressively shorter NTDs.

**Figure 2 cells-10-03441-f002:**
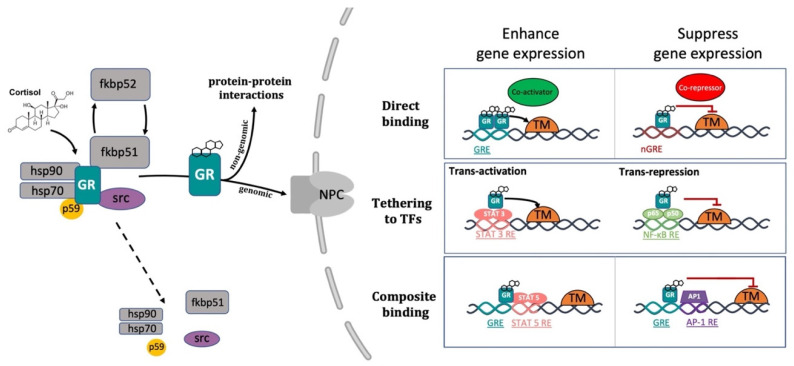
Representative picture of the canonical GR signalling pathway. After binding to GC, GR undergoes a FKBP51-FKBP52 mediated conformational change, becomes hyper-phosphorylated, dissociates from accessory proteins (chaperone complex) and finally translocates into the nucleus. Here, after dimerization with other GRs, it regulates the transcription of target genes by binding to DNA. Interestingly, GR may enhance or repress transcription of target genes by directly binding to palindromic GC response elements (GRE), or by tethering itself to other transcription factors apart from DNA binding, or in a composite manner by both directly binding GRE and interacting with transcription factors bound to neighbouring sites. Created with BioRender.com.

**Figure 3 cells-10-03441-f003:**
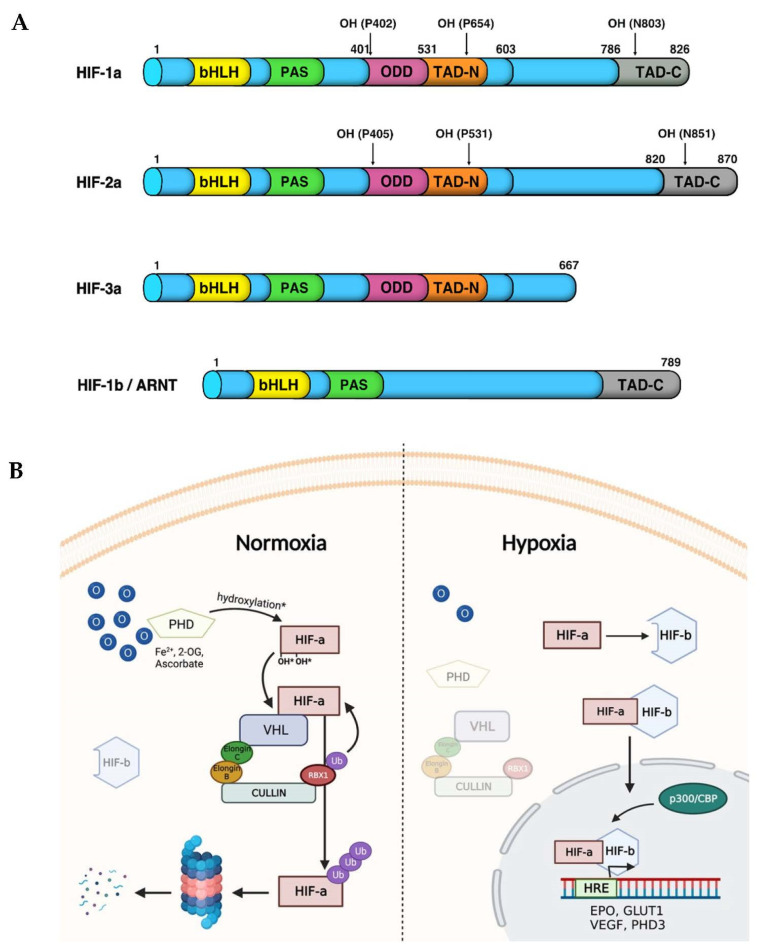
HIF-α isoforms and their receptor HIF-1β/ARNT structural domains. (**A**) In vertebrates both HIF-1α and HIF-2α, but not HIF-3α, contain a basic helix-loop-helix (bHLH) domain, a Per-Arnt-Sim domain (PAS), an oxygen dependent degradation (ODD) domain, an N-terminal transactivation domain (N-TAD) located in the ODD and a C-TAD localized in the C-terminal region. (**B**) The HIF signalling pathway. Under normoxic conditions, PHDs hydroxylate HIF-α subunits on two specific prolyl residues within the ODDD. In turn, VHL recognizes and binds to hydroxylated HIF-α and then recruits the other components of the E3-ubiquitin ligase complex. The latter promotes the ubiquitin-mediated proteasomal degradation of HIF-α subunits. Conversely, hypoxic conditions inhibit PHD activity, and the subsequent degradation of HIF-α, which can, in turn, be stabilized in the cytoplasm, can dimerize with HIF-β-subunit and migrate into the nucleus. Here, the HIF-αβ active complex enhance the expression of target genes such as PHD3, VEGF, GLUT1 and EPO involved in restoring oxygen homeostasis. Created with BioRender.com.

**Figure 4 cells-10-03441-f004:**
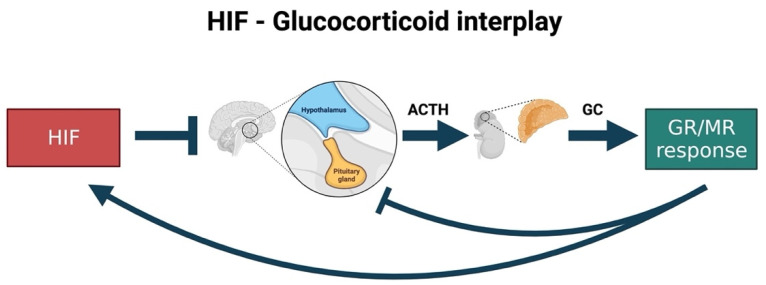
Speculative scheme of how putative HIF-GC crosstalk occurs in the zebrafish. Previous research highlighted that GCs may enhance both GR and MR-mediated responsiveness and stimulate the HIF signalling. Conversely, upregulated HIF levels may inhibit GC biosynthesis by negatively regulating *pomca* at the level of the pituitary gland. Consequently, this is believed to occur as a logic homeostatic response exerted by HIF signalling to control not only its own activity but that of glucocorticoids to facilitate inflammation resolution and homeostasis. Created with BioRender.com.

**Figure 5 cells-10-03441-f005:**
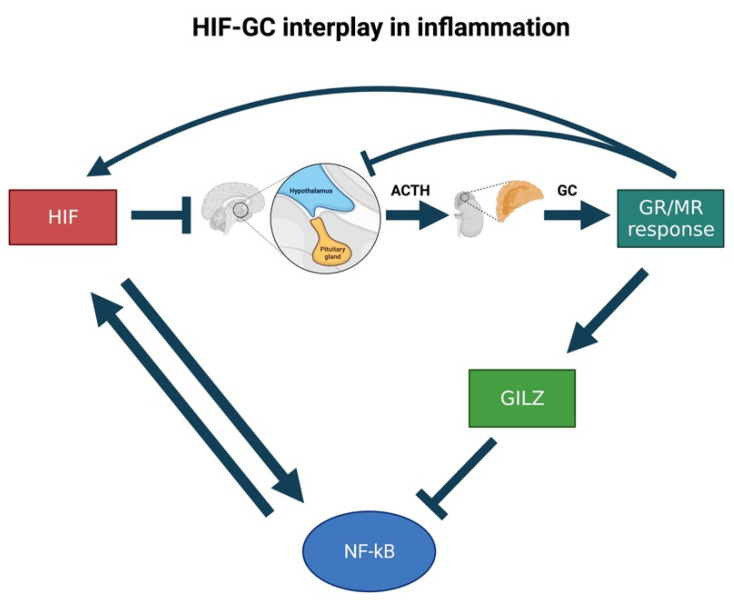
Speculative scheme of how the putative HIF-GC crosstalk occurs in inflammation. We speculate that HIF can negatively regulate *pomca* expression to control not only its own activity but also that of glucocorticoids throughout the inflammatory process. Indeed, if not properly controlled, glucocorticoids could upregulate GILZ, whose overexpression has been demonstrated to suppress NF-κB activation. Since the latter can, in turn, increase the transcription of HIF-1α in response to hypoxic and inflammatory stimuli, its downregulation would hamper HIF activation and avoid essential pro-inflammatory HIF-mediated effects to occur.

## Data Availability

Not applicable.

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
