# Peer review of "Homeostatic Regulation of Glucocorticoid Receptor Activity by Hypoxia-Inducible Factor 1: From Physiology to Clinic"

_cells, 2021, doi:10.3390/cells10123441_

Round 1

Reviewer 1 Report

I think the work in this review is comprehensively described in a concise manner and ends up in a viewpoint on putative crosstalk between HIF and glucocorticoid hormones. The latter hypothesis is an exciting contribution to the field that definitely needs clarification of the molecular mechanisms as it may help to explain the development of glucocorticoid resistance.

Clinical implications of this potential crosstalk are provided in this work and it will be interesting to see whether HIF overactivity in critically ill COVID-19 patients may contribute to adrenal insufficiency observed in these patients (in addition to e.g. direct pathogenic effects of SARS-CoV-2 on the components of the HPA-axis).

Minor comments

Line 69 ‘whereas GR under stressful conditions’ or at the diurnal peak

Line 366  Section 2.5 on the role of glucocorticoids in inflammation: (Line 377) ’MR-dependent pro-inflammatory effects have also been noted’. It should be added that it is presently unclear whether these effects are mediated by the glucocorticoid cortisol or the mineralocorticoid aldosterone (Cardiovascular Research, Volume 114, Issue 7, 01 June 2018, Pages 944–953).

Author Response

We  thank  the  Reviewer  for  his  positive  comments  and  careful  review,  which  helped improve the manuscript.

Reviewer 2 Report

I thought that the paper was well-done. It is very comprehensive and interesting. 

1. The authors review the interactions between the HPA axis and HIF-1. This is an excellent review about the function of both systems.
2. The topic has been covered but this is an updated review that provides new information about the interactions between glucocorticoid activity and HIF-1
3.  It adds the interaction between the two systems, compared with other published material.
4. The conclusions are consistent with the evidence and arguments presented and address the main question posed.
5. The references are very complete
6. The figures are fine.

Author Response

We thank the Reviewer for his positive comments. 

Reviewer 3 Report

Review report – Homeostatic regulation of glucocorticoid receptor activity by hypoxia-inducible factor 1: from physiology to clinic

  1. A brief summary (one short paragraph) outlining the aim of the paper, its main contributions and strengths.

In this review, the authors provide a profound overview of the crosstalk between glucocorticoid (GC) signalling and hypoxia. They give an elaborate introduction of both signalling pathways before the authors describe the involvement of GCs in inflammation and how a crosstalk between hypoxia-inducible factors (HIFs) and GCs is present. The review focusses on what is already known based on in vitro studies and in vivo experiments using zebrafish larvae. They finish with the involvement of HIF proteins in the GC biosynthesis and responsiveness in zebrafish.

  1. General concept comments

Review: commenting on the completeness of the review topic covered, the relevance of the review topic, the gap in knowledge identified, the appropriateness of references, etc.

The review is well written and provides a clear state of the art of the literature. The authors also added some nice figures as illustration. They also provide a good overview of what is known about the interaction between the glucocorticoid receptor (GR) and HIF in zebrafish, which is a nice addition to the recently published review https://doi.org/10.3389/fimmu.2021.684085.

I only have a few minor comments:

It would be nice to add a figure considering the role of GCs and hypoxia in inflammation in the corresponding subset of the review, how we have to see these interactions. Also based on the in vitro studies where the crosstalk between the glucocorticoid receptor (GR) and HIF is investigated.

In the introduction of the review, the authors mention the oxygen labile HIF-α isoforms, however these proteins itself are not sensitive to oxygen levels, these are the PHD proteins which are involved in the degradation of the HIF proteins. Furthermore, it would be good to add what also regulates the PHD activity next to the oxygen levels both in the manuscript and in Figure 3.

It would be useful to make a clear distinction in the text when referring to results in human and zebrafish e.g. the classical GRE in human is AGAACAnnnTGTTCT compared to GGAACAnnnTGTTCT in zebrafish (line 293), similar with PHD3 (line 113-114) in zebrafish compared to PHD2 in human and mice.

In figure 2: do the authors mean GR dimerization instead of tetramerization?

Considering the role of GCs in inflammation, the authors might want to include a recently published paper describing the requirement of DNA binding for the anti-inflammatory function of the GR: https://doi.org/10.1093/nar/gkaa565

In the third part where the authors describe the HIF signalling pathway, line 449-459 is a repetition of the text mentioned in the introduction of the review. The authors might consider to shorten this at the beginning of review? Also in this part of the review, it might be good that the authors shortly describe the different domains present in HIF proteins and what their function is.

When the authors describe the role of hypoxia in inflammation, they mention the attenuation of the anti-inflammatory effect of the GR by hypoxia (line 530-531). It might be useful to add the recently published paper showing the reduced anti-inflammatory capacity of GR in mice https://doi.org/10.15252/embr.202153083

Considering the in vitro studies showing the interaction between GR and HIF, the authors might want to add this paper: https://doi.org/10.1016/j.mce.2020.111007

General questions to help guide your review report for review articles

  • Is the review clear, comprehensive and of relevance to the field?

Yes

  • Is a gap in knowledge identified?

No

  • Was a similar review published recently and, if yes, is this current review still relevant and of interest to the scientific community?

Yes, the recently published review https://doi.org/10.3389/fimmu.2021.684085, but the authors created an added value by adding literature considering the crosstalk between GR and HIF in zebrafish and how the mineralocorticoid receptor (MR) might be involved

  • Are the cited references current (mostly within the last 5 years)? Are any citations omitted? Does it include an abnormal number of self-citations?

No comments on the references used in this review

  • Are the statements and conclusions drawn coherent and supported by the listed citations?

Yes

  • Are the figures/tables/images/schemes appropriate? Do they properly show the data? Are they easy to interpret and understand?

Yes
